# Decarboxylative thiolation of redox-active esters to free thiols and further diversification

Tianpeng Cao[1], Tianxiao Xu[1], Ruting Xu[1], Xianli Shu[1] & Saihu Liao [1,2✉]

Thiols are important precursors for the synthesis of a variety of pharmaceutically important sulfur-containing compounds. In view of the versatile reactivity of free thiols, here we report the development of a visible light-mediated direct decarboxylative thiolation reaction of alkyl redox-active esters to free thiols based on the abundant carboxylic acid feedstock. This transformation is applicable to various carboxylic acids, including primary, secondary, and tertiary acids as well as natural products and drugs, forging a general and facile access to free thiols with diverse structures. Moreover, the direct access to free thiols affords an advantage of rapid in situ diversification with high efficiency to other important thiol derivatives such as sulfide, disulfide, thiocyanide, thioselenide, etc.

[1] Key Laboratory of Molecule Synthesis and Function Discovery (Fujian Province University), College of Chemistry, Fuzhou University, 350108 Fuzhou, China. [2] State Key Laboratory of Photocatalysis on Energy and Environment, College of Chemistry, Fuzhou University, 350108 Fuzhou, China. ✉email: shliao@fzu.edu.cn

The construction of molecule libraries with structural and functional diversity is crucial for the study in the context of chemical biology and drug discovery[1–3]. Thiols are important precursors for the synthesis of a variety of pharmaceutically important sulfur-containing compounds, including sulfonamides, sulfonyl fluorides, sulfoxides, sulfides, disulfides, and so on, by virtue of their high reactivity and valence labile nature, and widely employed in organic synthesis, polymer preparation, materials science, and biomedicine[4–17]. In fact, besides their well-known roles in protein structure stabilizations[18,19] and many enzymatic processes[20], thiol is also one of the most targeted sites in post-translational protein modification (Fig. 1a)[21–23]. Inspired by the versatile reactivity of thiols, we conceived that, based on the feedstock of abundant carboxylic acid, a decarboxylative thiolation of acid-derived redox-active esters (RAEs)[24–39] (RCO₂A*) to free thiols could forge a novel access to various thiols and related derivatives with considerable structural diversity. In particular, the decarboxylative access to free thiols could allow a further diversification to other sulfur-containing compounds[40–42] with a multiplied diversity by varying the coupling agents (e.g., with various electrophiles E⁺, Fig. 1b).

A number of radical C–S bond formation reactions[43–49] have been reported, including the related decarboxylative transformations pioneered by Barton in 1980s[46–49], but a direct radical thiolation to free thiols remains elusive so far. The challenges for the proposed radical decarboxylative thiolation to free thiols probably lie in the labile nature of free thiols, which can lead to dimerization, undesired hydrogen transfer, and other side reactions[43]. In fact, free thiols are commonly used as hydrogen atom transfer (HAT) catalysts or reagents in radical chemistry[50–54], and the HAT from a primary alkyl thiol to alkyl radicals is a fast process (ca. $10^7 \, M^{-1} s^{-1}$)[52–54]. Therefore, in the decarboxylative thiolation process, the desired thiol product (RSH) formed earlier may intercept the newly generated alkyl radicals (R·), thus leading to the undesired alkane (R-H) formation (Fig. 1c). Nevertheless, in radical polymerization, the chain-transfer agents (CTA) employed in reversible addition-fragmentation chain-transfer polymerization can readily alter the radical addition rate by adjusting the Z group and increase $k_{add}$ to above $10^8 \, M^{-1} s^{-1}$ (Fig. 1, C, below)[55,56], which inspired us to focus on the sulfur donor search in the beginning. Herein, we report our efforts in the successful identification of aryl thioamides as an effective sulfur donor, and the invention of visible light-mediated direct

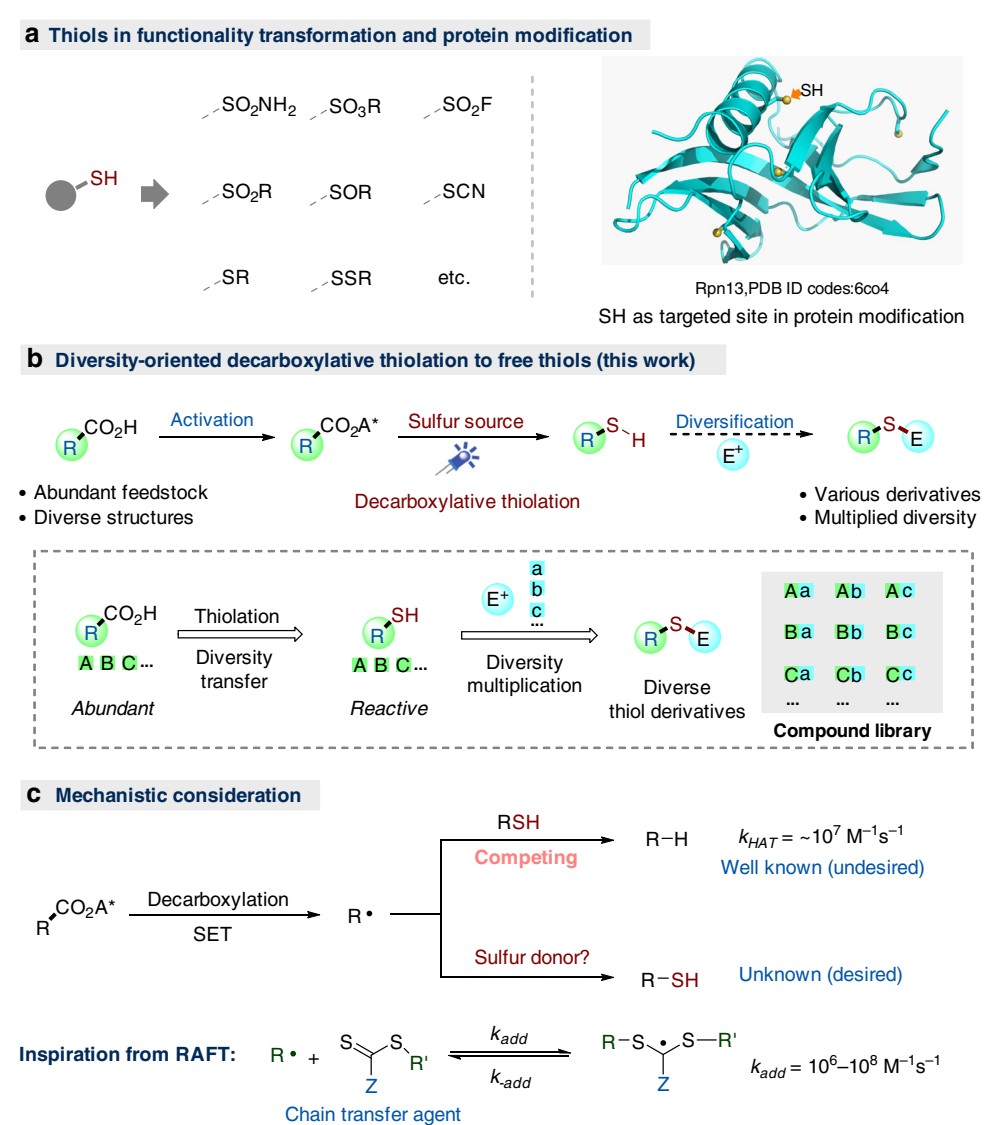

**Fig. 1 Reaction design. a** Thiols in functionality transformation and protein modification. **b** Diversity-oriented decarboxylative thiolation to free thiols. **c** Mechanistic consideration and the inspiration from reversible addition-fragmentation chain-transfer (RAFT) polymerization.

decarboxylative thiolation of alkyl RAEs to free thiols. Moreover, further diversification to other thiol derivatives, such as sulfide, disulfide, thiocyanide, and thioselenide via in situ trapping, is also demonstrated.

## Results

**Reaction optimization.** We commenced our study with the search for sulfur donors suitable for the radical thiolation reaction, by employing dihydrocinnamic acid-derived RAE (**1**) as the model substrate and Eosin Y-Na$_2$/diisopropylethylamine (DIPEA) as the photoredox catalytic system (Table 1). Initially, thiourea **2a**, which is frequently used as a sulfur donor in the nucleophilic substitution reactions of alkyl halides[57,58], was examined first in the reaction, but only the alkane product **3'** was observed (entry 1), indicating the radical reactivity is substantially different from the polar substitution reactions. Other thioureas like **2b** and **2c** were also examined, but neither of them afforded the desired thiol product (entries 2 and 3). We then turned our attention to other types of sulfur donor (for more details about the reaction development, please see the Supplementary Figs. 1–9 and Table 1). To our delight, ben-zothioamide was found being a promising sulfur donor for this decarboxylative thiolation reaction, and the desired thiol **3** could be obtained as the predominant product in 77% yield (entry 4). We then carried out several modifications on the phenyl group of benzothioamides (entries 4–6). Electron-withdrawing group (-CF$_3$, **2e**) led to a decreased yield of 35%, while the introduction of an electron-donating group (**2f**) was

found beneficial and further increased the yield to 81% (entry 5 vs. entry 6). The N-H group proved to be crucial for this transformation. Replacement with either one or two methyl groups (**2g** and **2h**), both resulted in a sharp drop in yield (entries 7 and 8). Moreover, sulfur powder was also tested, but no desired thiol product was observed (entry 9). With **2f** as the sulfur donor, we conducted a further reaction optimization, including

photocatalyst, solvent, light source, and so on (for details, please see the Supplementary Tables 2 and 3). Other photo-catalysts, such as Ru(bpy)$_3$Cl$_2$·6H$_2$O and Ir(ppy)$_3$, gave lower yields (entries 10–13), while Eosin Y was found equally efficient (entry 14). Running the reaction in CH$_3$CN could slightly enhanced the selectivity (entry 15). Without light or photo-catalyst, no reaction or a low yield was observed (entries 16 and 17). To our delight, the employment of two equivalents of sulfur donor **2f** could further suppress the undesired alkane formation and increase the yield of the desired thiol product to a decent level of 88% in the end (entry 18).

**Substrate scope.** With the optimized reaction conditions in hand, we next examined the reaction scope with a variety of primary, secondary, and tertiary acid-derived RAEs (Fig. 2). Some free thiols are volatile and thus isolated in their disulfide form by in situ trapping with diphenyl disulfide. These results are also included in Fig. 2. In cases of primary acids (**3–18**), we could see a good functional group tolerance. Br, Cl, ether, ester, and also a triple C–C bond are all compatible in the reaction, and the

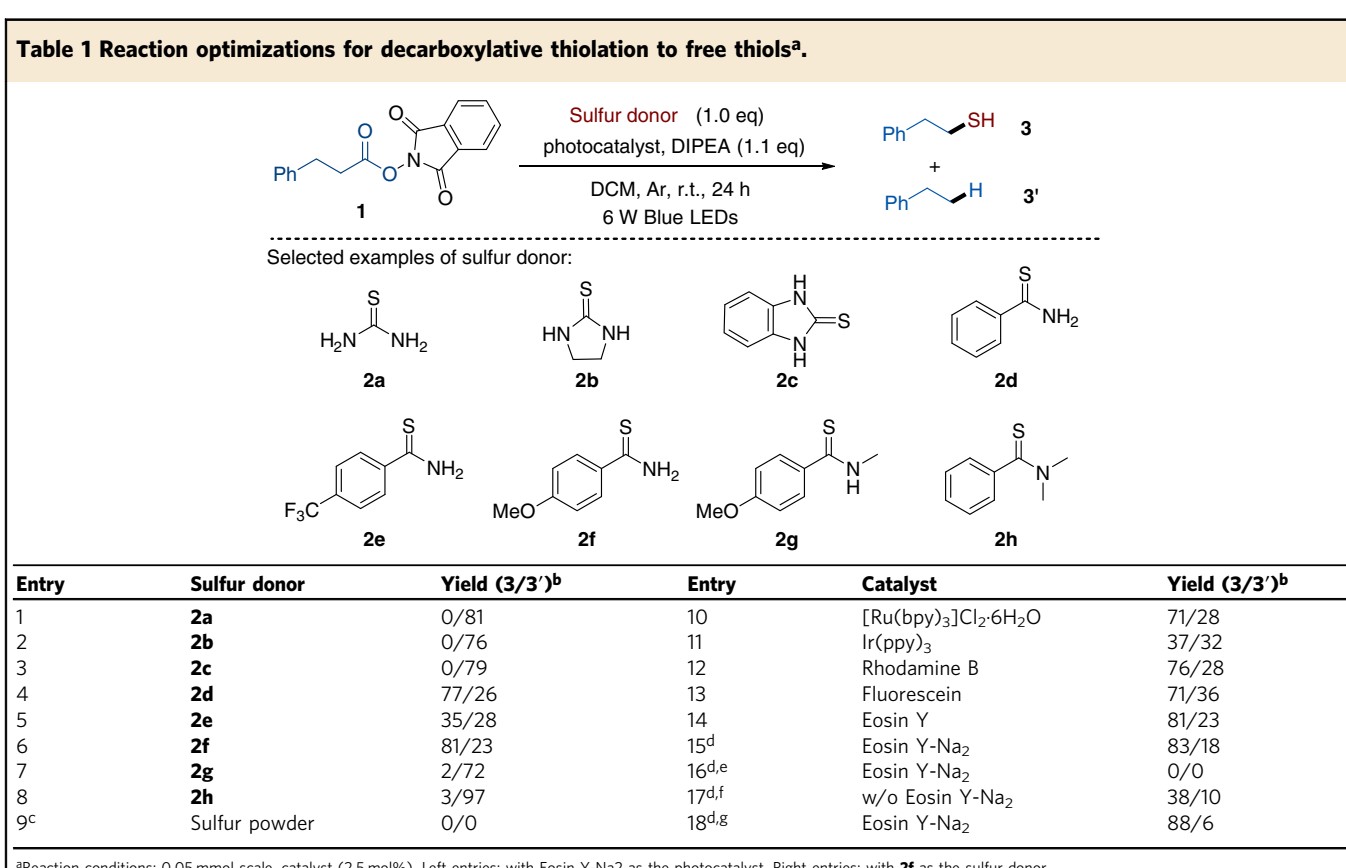

**Table 1 Reaction optimizations for decarboxylative thiolation to free thiols[a].**

| Entry | Sulfur donor | Yield (3/3')[b] | Entry | Catalyst | Yield (3/3')[b] |
|---|---|---|---|---|---|
| 1 | **2a** | 0/81 | 10 | [Ru(bpy)$_3$]Cl$_2$·6H$_2$O | 71/28 |
| 2 | **2b** | 0/76 | 11 | Ir(ppy)$_3$ | 37/32 |
| 3 | **2c** | 0/79 | 12 | Rhodamine B | 76/28 |
| 4 | **2d** | 77/26 | 13 | Fluorescein | 71/36 |
| 5 | **2e** | 35/28 | 14 | Eosin Y | 81/23 |
| 6 | **2f** | 81/23 | 15[d] | Eosin Y-Na$_2$ | 83/18 |
| 7 | **2g** | 2/72 | 16[d,e] | Eosin Y-Na$_2$ | 0/0 |
| 8 | **2h** | 3/97 | 17[d,f] | w/o Eosin Y-Na$_2$ | 38/10 |
| 9[c] | Sulfur powder | 0/0 | 18[d,g] | Eosin Y-Na$_2$ | 88/6 |

[a]Reaction conditions: 0.05 mmol scale, catalyst (2.5 mol%). Left entries: with Eosin Y-Na2 as the photocatalyst. Right entries: with **2f** as the sulfur donor.
[b]Determined by GC-MS analysis with anisole as an internal standard.
[c]Sulfur powder (5.0 equiv.).
[d]Reaction was performed in CH$_3$CN instead of DCM.
[e]In dark.
[f]Without photocatalyst.
[g]With 2.0 equiv. of **2f**.

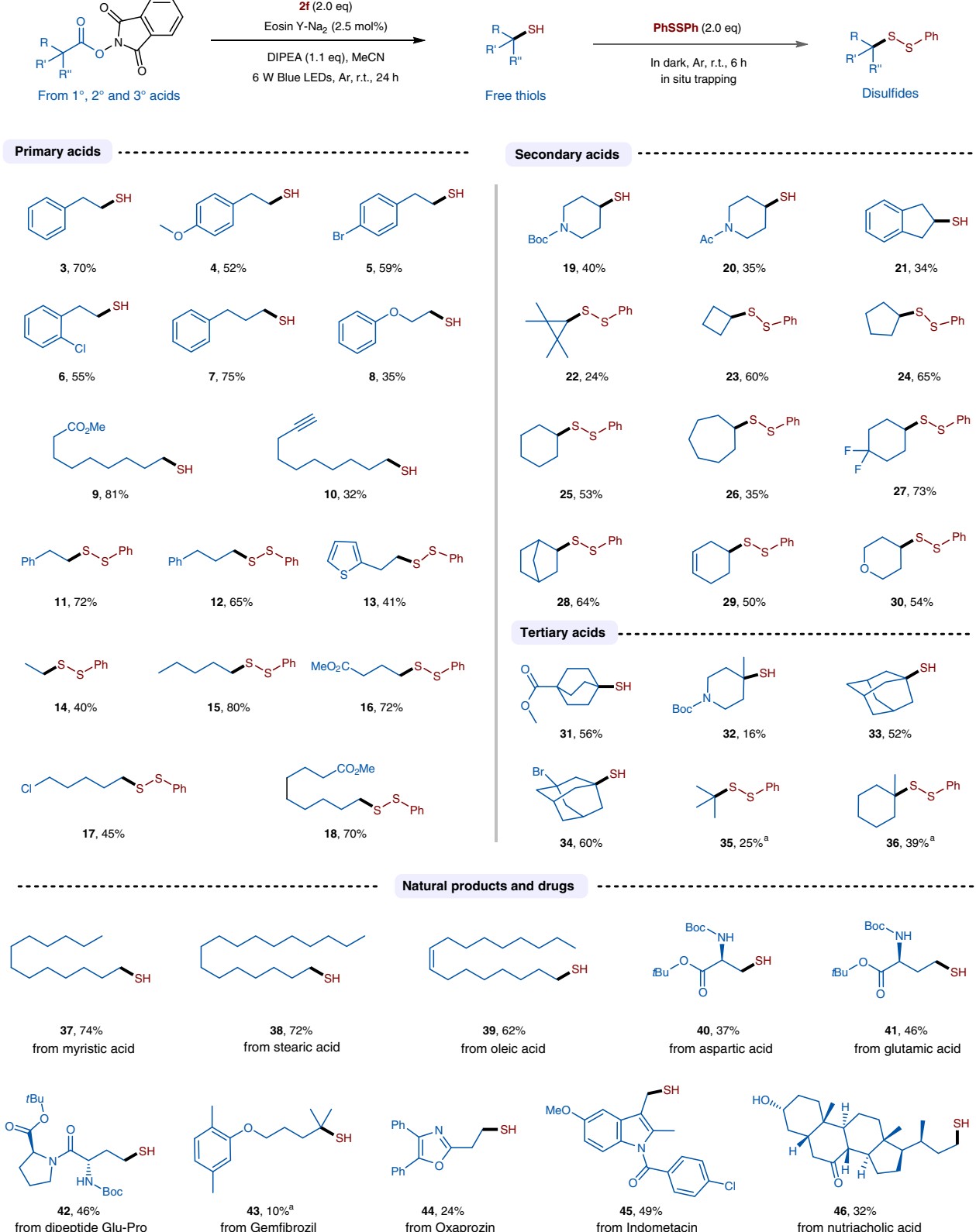

**Fig. 2 Substrate scope.** Reactions were performed on a 0.2 mmol scale, and trapping reactions were conducted with 2.0 equiv. of PhSSPh in dark for 6 h. [a]In CF$_3$CH$_2$OH.

desired free thiols (**4–10**) or disulfides (**11–18**) could be isolated in moderate to good yields. Through this decarboxylative thiolation, simple propionic acid can also be converted to the corresponding disulfide in 40% yield (**14**). The reaction can be well applied to cyclic carboxylic acids (**19–30**), and different ring sizes (**22–26**), including cyclopropane (**22**), cyclobutane (**23**), benzocyclopentane (**21**), and cycloheptane (**26**), all can be converted to the desired product in reasonable yields. Heterocyclic free thiols

or disulfides, such as **19**, **20**, and **30**, are also accessible from the corresponding carboxylic acids. Pleasingly, tertiary thiols or disulfide (**31–36**) can be prepared as well with this method. In particular, this visible light-mediated radical decarboxylative thiolation reaction could afford a facile access to tertiary bridgehead thiols (such as bicyclic thiol **31**, 1-adamantanethiol **33** and **34**) under mild reaction conditions, which are often difficult to prepare via traditional nucleophilic substitution reactions due to the steric shielding of the bridgehead position that prevents the backside attack of the nucleophiles[59]. The reported synthesis of **33** from 1-adamantyl bromide or alcohol was performed under very harsh conditions (reflux in AcOH/conc. aq. HBr), and under the same conditions, only a trace amount of product was obtained in the synthesis of bicyclo[2.2.2] octane-1-thiol[59,60]. Moreover, the decarboxylative thiolation reaction can be well extended to natural occurring acids, such as oleic acid (**39**), aspartic acid (**40**), glutamic acid (**41**), dipeptide Glu-Pro (**42**), and nutriacholic acid (**46**). It is worth mentioning that the conversion of aspartic acid to the thiol product **40** is resembling a transformation of aspartic acid to cysteine via a residue manipulation. To our delight, this decarboxylative thiolation can be adopted for the late-stage modification of drugs, such as gemfibrozil (**43**), oxaprozin (**44**), and indometacin (**45**). In cases of **35**, **36**, and **43**, we could observe less alkane formation when the reactions were performed in $CF_3CH_2OH$ instead of $CH_3CN$, and thus led to a better yield. As outlined in Fig. 2, under this reaction condition, primary, secondary, and tertiary carbon radicals can all be readily generated from the corresponding alkyl NHPI esters and subsequently trapped by the sulfur donor and converted to the desired thiol or disulfide products.

**Product diversification**. Disulfides are important motifs in life and biological active molecules, due to their unique pharmacological and physiochemical properties[61–66]. To further explore the scope of this transformation, more diaryl disulfide-trapping agents were examined with dihydrocinnamic acid-derived ester **1** as a model substrate. As shown in Fig. 3a, the reactions worked well with various diaryl disulfides processing different electronic nature, affording the desired unsymmetric alkyl aryl disulfides **47–51** readily in good yields. Importantly, the conversion of carboxylic acids to disulfides via this dicarboxylic thiolation/in situ trapping protocol provides a facile approach to synthesize this type of molecules with high structural diversity. Thiols are key precursors to many pharmaceutically important compounds[4–17,40–45]. The direct decarboxylative thiolation to free thiols allows for the establishment of a rapid, in situ diversification to various thiol derivatives without isolating the free thiols, which are often smelly and unstable. As shown in Fig. 3b, the 2-phenylethane-1-thiol can readily undergo alkylation in situ with various electrophiles or Michael acceptors, to provide the corresponding sulfide products (**52–57**). Moreover, trapping with 4-methylbenzenesulfonyl cyanide enable the conversion of carboxylic acid to thiocyanide **58**. An access to thioselenide from the corresponding carboxylic acid via an in situ reaction with diphenyl diselenide was also demonstrated with the synthesis of thioselenide **59**.

A possible reaction mechanism is proposed as outlined in Fig. 4. Under the irradiation of light, the photocatalyst Eosin Y (**PC**) is excited and subsequently reductively quenched by DIPEA or **2f**, affording **PC•−** and **2f′** in the presence of base. The fluorescence quenching experiments also clearly showed that both DIPEA and sulfur donor **2f** can quench the fluorescence of the photocatalyst (Supplementary Figs. 7–9). A single electron transfer (SET) from **PC•−** to the carboxylic acid-derived RAE afford the corresponding radical anion **Int-A** and concurrently regenerate the photocatalyst (Path A). As product also observed in the absence of **PC** and **2f** showed substantial absorption in the blue light region (Supplementary Fig. 5), Path B might also

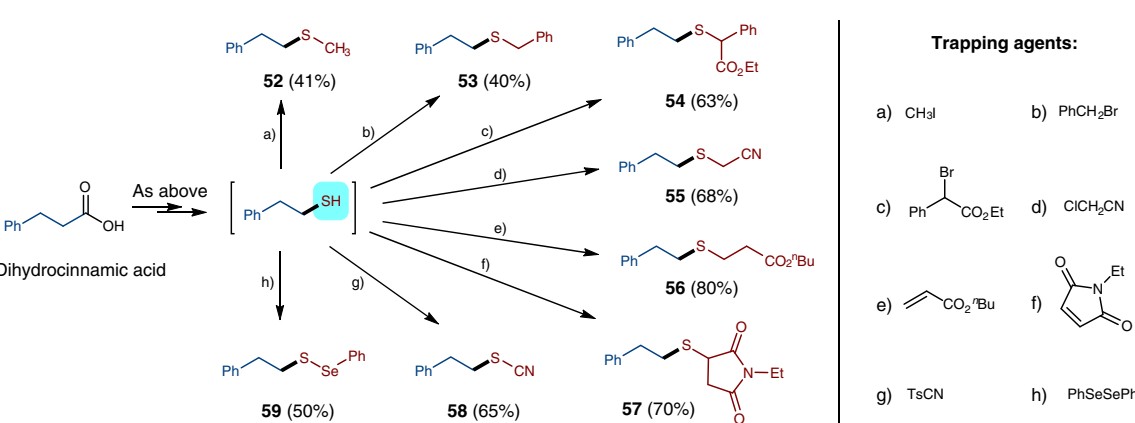

**Fig. 3 Reaction extension and product diversification. a** Extension to disulfide synthesis. **b** In situ diversification.

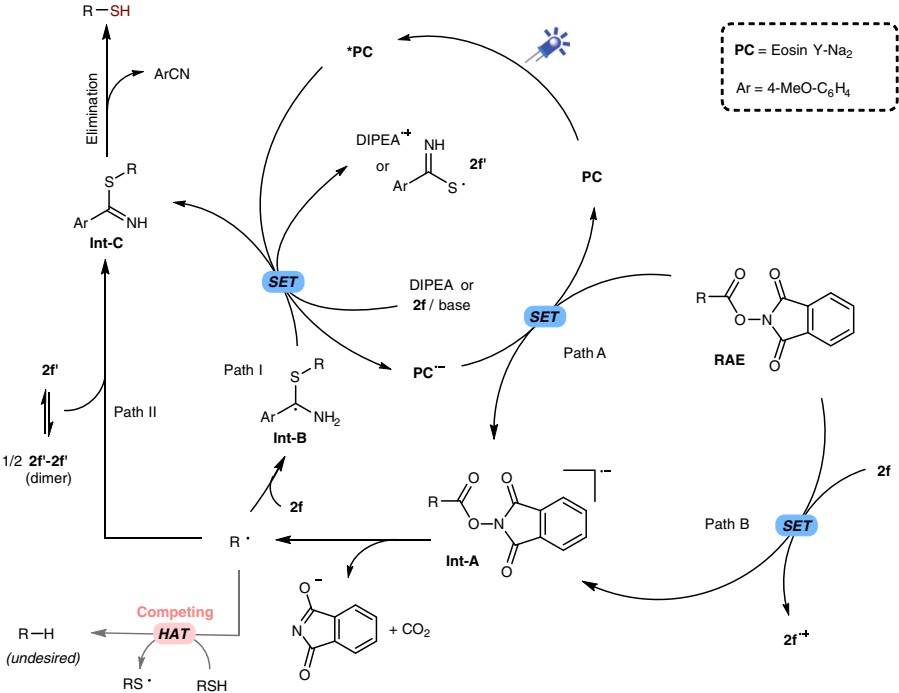

**Fig. 4 Reaction pathways.** A mechanistic proposal for the decarboxylative thiolation of redox-active esters (RAEs) to free thiols. SET single electron transfer, HAT hydrogen atom transfer.

involve to some extent. **Int-A** then undergoes fragmentation to give the alkyl radical **R·** via the N–O bond cleavage followed by the extrusion of $CO_2$. The alkyl radical can be trapped with TEMPO (Supplementary Fig. 3). The addition of **R·** to the sulfur donor **2f** generate the radical intermediate **Int-B**[67], which can be oxidized to imine **Int-C** by a SET to excited photocatalyst as shown as Path I. The higher yields with aryl thioamides than alkyl ones may be ascribed to the stabilization of the aryl group to radical **Int-B**. The beneficial effect of the electron-donating group (**2d–f**, entries 4–5, Table 1) might result from a favored one electron oxidation of **Int-B** to **Int-C**. Alternatively, a radical coupling of **R·** and **2 f'** could also afford the intermediate **Int-C** (Path II). As a competing process, the HAT from thiol (R-SH) to **R·** will lead to the formation of the alkane side product (R–H)[51,52], which can be suppressed by a rapid trapping of **R·** with active thiolating agents via Path I or II. In the end, the desired thiol is produced after the elimination of one molecule of ArCN[68,69], which could be a relatively slow step, and a slow release of free thiols can decrease the formation of the undesired alkane product. The nitrile byproduct formation was confirmed by gas chromatography-mass spectrometry (GC-MS) analysis, and it can also be isolated by column chromatography. Therefore, the use of N-unsubstituted thioamides is crucial for this transformation. In contrast, N-substituted thioamide **2g** and **2h** are unable to form the corresponding N-H imine intermediate **Int-C**.

## Discussion

In conclusion, a visible light-mediated direct decarboxylative thiolation of carboxylic acid-derived RAEs to free thiols has been developed. Aryl thioamides have been identified as an effective sulfur donor and crucial to this thiolation reaction. The transformation of abundant carboxylic acid feedstock to the corresponding free thiols and their further in situ diversification allows for a rapid and general access to various pharmaceutically important compounds, such as sulfide, disulfide, thiocyanide, and thioselenide, with diverse structures, which may be utilized for the molecule library construction and benefit the related study in chemical biology and discovery of novel, biologically interesting small molecules.

## Methods

**General procedure for decarboxylative thiolation**. To an oven-dried 10-ml Schlenk tube equipped with a magnetic stir bar and a Teflon-coated septum screwcap was added the NHPI redox-active ester (0.2 mmol, 1.0 equiv.), 4-methoxythiobenzamide (**2f**, 0.4 mmol, 2.0 equiv.), Eosin Y-Na$_2$ (2.5 mol%). The tube was evacuated and backfilled with argon for three cycles. The DIPEA (0.22 mmol, 1.1 equiv.) and dry CH$_3$CN (2.0 ml) was added via a gastight syringe under argon atmosphere. Make sure the screwcap was closed, and the solvent was frozen by liquid nitrogen. Then, the screwcap was opened and the tube was evacuated for about 3 min. The screwcap was closed and let the solvent melts in a tepid water bath. Repeat above freeze-pump-thaw procedures for 3–5 times until you no longer see the evolution of gas as the solution thaws. The tube was filled with argon and sealed, irradiated with 6 W blue light-emitting diode (LED) reactor and stirred at ambient temperature for 24 h. Full experimental details (Supplementary Figs. 1–9 and Supplementary Tables 1–3) and characterization of new compounds (Supplementary Figs. 10–68) can be found in the Supplementary Methods section.

**General procedure for the synthesis of disulfides**. To an oven-dried 10-ml Schlenk tube equipped with a magnetic stir bar and a Teflon-coated septum screwcap was added the NHPI redox-active ester (0.2 mmol, 1.0 equiv.), 4-methoxythiobenzamide (**2f**, 0.4 mmol, 2.0 equiv.), and Eosin Y-Na$_2$ (2.5 mol%). The tube was evacuated and backfilled with argon for three cycles. The DIPEA (0.22 mmol, 1.1 equiv.) and dry CH$_3$CN (2.0 ml) was added via a gastight syringe under argon atmosphere. Make sure the screwcap was closed, and the solvent was frozen by liquid nitrogen. Then, the screwcap was opened and the tube was evacuated for ~3 min. The screwcap was then closed and the solvents were let to melt in a tepid water bath. Repeat above freeze-pump-thaw procedures for 3–5 times until you no longer see the evolution of gas as the solution thaws. The tube was filled with argon and sealed, irradiated with 6 W blue LED reactor and stirred at ambient temperature for 24 h. Then, the K$_2$CO$_3$ (0.4 mmol, 2.0 equiv.) and diaryl disulfide (0.4 mmol, 2.0 equiv.) was added under argon atmosphere. The tube stirred at ambient temperature for 6 h in the dark. Upon completion, the reaction mixture was carefully concentrated and the residue was further purified by flash chromatography to give the desired disulfide products. Full experimental details and characterization of new compounds can be found in the Supplementary Methods section.

## Data availability
The authors declare that all data supporting the findings of this study are available within the article and Supplementary information files, and are also available from the corresponding author upon reasonable request.

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

## Acknowledgements

We gratefully acknowledge National Natural Science Foundation of China (No. 21602028), the Recruitment Program of Global Experts, and Fuzhou University for the financial support.

## Author contributions

T.C. and T.X. contributed equally to this work. T.C. developed the reactions, and contributed to the reaction scope investigation, mechanistic study, and product derivatization. T.X. carried out most substrate synthesis, and contributed to the study of reaction scope, product diversification, and reaction mechanism. R.X. and X.S. participated in the synthesis of substrates. S.L. conceived this concept and prepared this manuscript with feedback from T.C. and T.X.

## Competing interests

The authors declare no competing interests.
