## [Peer Review File · Nature Communications]

REVIEWER COMMENTS

Reviewer #1 (Remarks to the Author):

Liao and co-workers described the invention of a visible light-mediated decarboxylative thiolation of alkyl redox-active esters to free thiols under the concept of diversity-oriented synthesis. The decarboxylative access to free thiols under a photoredox radical conditions is a challenging reaction. The optimization of the reaction was focused on choice of sulfur donor, and good yield was finally achieved under mild reaction conditions by suppressing the competing formation of hydrodecarboxylation products. The reaction scope is wide with primary, secondary, and tertiary acids, and the yields are reasonable good. The transformation of natural carboxylic acids and drugs to thiols was also demonstrated. The authors further extended the method to the synthesis of disulfides, sulfides, thiocyanide, thioselenide via in-situ trapping of the free thiols. Based on control experiments, fluorescence titrations, and key intermediate trappings, a plausible mechanism was proposed.

The decarboxylative thiolation of alkyl redox-active esters to free thiols is new. In fact, the transformations of carboxylic acids to disulfides, thiocyanide, and thioselenide etc. demonstrated in this paper are also new methods. In my opinion, this work can attract considerable interest as 1) the visible-light mediated decarboxylative transformation is a very hot research area, and 2) the method affords a new and useful method for the synthesis of various important sulfur-containing compounds with diverse structure, which could be of interest to labs of chemical biology and drug discovery.

Overall, this work represents a new and significant contribution to both areas of photo-induced decarboxylative transformations and thiol/thiol-derivative synthesis. The authors lay out a well-organized paper that defines the challenges, significances, and advances clearly, and the conclusion is well supported by the results. I am therefore supportive of publishing this paper in Nature communication after minor revisions.

1) Figure 1A, in the structure of Rpn13, the thiol group (-SH) is disconnected from the protein. It can be shown as yellow dot (?): -SH, as I saw there are several thiol residues in this protein.

2) In the substrate scope, reactions of 34 and 40 were performed in CF₃CH₂OH rather than CH₃CN, any comment on choosing this solvent?

3) The reviewer believes that the novelty and the establishment of this transformation could already ensure this paper to be published in a top journal, but still curious about the mechanism: a) what happened without adding DIPEA or with an inorganic base instead of DIPEA? b) how fast is the trapping by sulfur donor or active sulfur-species? May run a reaction with acrylates for comparison, which are often used as Michael acceptors in radical addition reactions.

4) This method provides a new and diversity-oriented method for the synthesis of thiols and also disulfides, which are very important compounds in both organic synthesis and drug discovery. In particular, the method offers an access to some molecules (e.g. tertiary thiols and disulfides) which otherwise would be difficult to prepare through conventional methods. One related paper published very recently may be cited appropriately: Li et al. Catalytic Decarboxylative Radical Sulfonylation, *Chem* 6, 1149-1159 (2020).

Reviewer #2 (Remarks to the Author):

In this manuscript Liao and co-workers report a method for the conversion of hydroxyphthalimide-derived alkyl esters to alkyl thiols. The reaction is promoted by a simple photoredox catalysts (Eosin Y) using blue LED irradiation and benzothioamide sulfur donor. A series of simple alkyl esters are demonstrated to undergo the reaction. Esther, aryl bromide, terminal alkyne, and

internal alkenes are tolerated in the reaction. The product alkyl thiols can be easily converted into other sulfur containing compounds by established reactions. A reasonable mechanism is proposed based on preliminary quenching studies. The manuscript provides a potentially useful addition to the growing class of redox-active ester decarboxylative thiolation reactions. Access to alkyl thiols without deprotection may have value to end-users in certain cases.

The manuscript title, abstract, and introduction discuss the reaction in the context of "diversity-oriented synthesis" and drug discovery. The paper describes a functional group interconversion on relatively simple substrates. In my opinion there is no direct connect to these fields, or at least not any more than prototypical functional group interconversions. The products generated have little structural diversity compared to modern pharmaceuticals or lead compounds. The introduction provides almost no context on the state-of-the-art in thiolation. The authors state SN2 reactions have limitations, however these limitations can be overcome by SN1 processes, epoxide opening reactions, Mitsunobu reactions, or conjugate thiolations (among others). In this light, the introduction, framing of the work, and referencing should be significantly revised.

In terms of the merit of the work with respect to thiolation and decarboxylation methodology, my opinion is the work is not suitable for publication in Nature Communications. The scope of substrates examined is narrow. Poor yields for tertiary substrates are observed for non-biased substrates. The strategy for acid activation and capture is well established. Due to the technical nature of the advance this manuscript reports, it is better suited for publication in a journal such as Org. Lett. or Chem. Eur. J.

Reviewer #3 (Remarks to the Author):

In this work, Liao and coworkers developed a visible light-mediated direct decarboxylative thiolation reaction of alkyl redox active esters to free thiols is developed based on the abundant carboxylic acid feedstock, and the arylthioamides have been identified as an effective sulfur donor and crucial to this thiolation reaction. Importantly, this transformation provides a convenient and green channel for the preparation of free thiols and could be further in-situ diversification allows for a rapid and general access to various pharmaceutically compounds such as sulfide, disulfide, thiocyanide, and thioselenide. However, there are still some issues that need to be resolved before being published. Therefore, I recommend this manuscript publish in nature communication after minor revised are needed.

1. For different activities of sulfur donors, the author should give corresponding explanations.
2. In Figure 2, the yields of some free thiols products are relatively low, what's the main byproduct?
3. If the carboxylic acid is used directly as the starting material, the corresponding product can be obtained by decarboxylation thiolation, or can the "one-pot" method be used to achieve this transformation?
4. Based on the speculated reaction mechanism, which step is the rate-determining step?
5. The "RSC Adv. 2016,6, 70335" should be cited in mechanism proposed.
6. The "J. Org. Chem. 1962, 27, 93-95" should be cited in introduction.
7. The figure S 7-9 should be provided the stern-volmer plots.

Dear Reviewers,

We have revised the manuscript and supplementary information accordingly. Thank you very much for the comments and suggestions. Below are our point-by-point responses:

Reviewer #1:

Liao and co-workers described the invention of a visible light-mediated decarboxylative thiolation of alkyl redox-active esters to free thiols under the concept of diversity-oriented synthesis. The decarboxylative access to free thiols under a photoredox radical conditions is a challenging reaction. The optimization of the reaction was focused on choice of sulfur donor, and good yield was finally achieved under mild reaction conditions by suppressing the competing formation of hydrodecarboxylation products. The reaction scope is wide with primary, secondary, and tertiary acids, and the yields are reasonable good. The transformation of natural carboxylic acids and drugs to thiols was also demonstrated. The authors further extended the method to the synthesis of disulfides, sulfides, thiocyanide, thioselenide via in-situ trapping of the free thiols. Based on control experiments, fluorescence titrations, and key intermediate trappings, a plausible mechanism was proposed.

The decarboxylative thiolation of alkyl redox-active esters to free thiols is new. In fact, the transformations of carboxylic acids to disulfides, thiocyanide, and thioselenide etc. demonstrated in this paper are also new methods. In my opinion, this work can attract considerable interest as 1) the visible-light mediated decarboxylative transformation is a very hot research area, and 2) the method affords a new and useful method for the thesis of various important sulfur-containing compounds with diverse structure, which could be of interest to labs of chemical biology and drug discovery.

Overall, this work represents a new and significant contribution to both areas of photo-induced decarboxylative transformations and thiol/thiol-derivative synthesis. The authors lay out a well-organized paper that defines the challenges, significances, and advances clearly, and the conclusion is well supported by the results. I am therefore supportive of publishing this paper in Nature communication after minor revisions.

1) Figure 1A, in the structure of Rpn13, the thiol group (-SH) is disconnected from the protein. It can be shown as yellow dot (?): -SH, as I saw there are several thiol residues in this protein.

Response: Thank you very much for the comments and suggestion. We have added an arrow to indicate the thiol group. Please see the revised Fig. 1 A.

2) In the substrate scope, reactions of 34 and 40 were performed in CF₃CH₂OH rather than CH₃CN, any comment on choosing this solvent?

Response: For these cases (some tertiary acids), a higher thiol/alkane ratio were observed in CF₃CH₂OH than that in CH₃CN, and thus led to slightly better yields. However, in cases of primary and secondary carboxylic acids, using alcoholic solvents led to the some hydrolysis (alcoholysis) of redox active esters and decreased the yield. We have added a comment on these results in the revised manuscript.

3) The reviewer believes that the novelty and the establishment of this transformation could

already ensure this paper to be published in a top journal, but still curious about the mechanism: a) what happened without adding DIPEA or with an inorganic base instead of DIPEA? b) how fast is the trapping by sulfur donor or active sulfur-species? May run a reaction with acrylates for comparison, which are often used as Michael acceptors in radical addition reactions.

Response: Without adding DIPEA, lower yield was obtained (please see Supplementary Table 3). With an inorganic base, such as K_2CO_3 , the decarboxylative thiolation could also proceed, but with a lower yield (68%). We observed some disulfide formation. As suggested, we performed a competing experiment with butyl acrylate (2.0 equivalents) as shown below. We found **3** and compound **1** were obtained as the major products, and only a trace amount of compound **2** was detected by GC-MS, which suggests that the rate of trapping by sulfur donor is faster than this Michael acceptor in the radical addition step. We have added these results in the revised Supplementary Information (Supplementary Table 3 & Scheme 1). Many thanks for the suggestion.

4) This method provides a new and diversity-oriented method for the synthesis of thiols and also disulfides, which are very important compounds in both organic synthesis and drug discovery. In particular, the method offers an access to some molecules (e.g. tertiary thiols and disulfides) which otherwise would be difficult to prepare through conventional methods. One related paper published very recently may be cited appropriately: Li et al. Catalytic Decarboxylative Radical Sulfonylation, Chem 6, 1149-1159 (2020).

Response: We have cited this paper as Ref. 17 in the revised manuscript. Thank you very much again for all the comments and suggestions.

Reviewer #2:

In this manuscript Liao and co-workers report a method for the conversion of hydroxyphthalimide-derived alkyl esters to alkyl thiols. The reaction is promoted by a simple photoredox catalysts (Eosin Y) using blue LED irradiation and benzothioamide sulfur donor. A series of simple alkyl esters are demonstrated to undergo the reaction. Ester, aryl bromide, terminal alkyne, and internal alkenes are tolerated in the reaction. The product alkyl thiols can be easily converted into other sulfur containing compounds by established reactions. A reasonable mechanism is proposed based on preliminary quenching studies. The manuscript provides a

potentially useful addition to the growing class of redox-active ester decarboxylative thiolation reactions. Access to alkyl thiols without deprotection may have value to end-users in certain cases.

The manuscript title, abstract, and introduction discuss the reaction in the context of “diversity-oriented synthesis” and drug discovery. The paper describes a functional group interconversion on relatively simple substrates. In my opinion there is no direct connect to these fields, or at least not any more than prototypical functional group interconversions. The products generated have little structural diversity compared to modern pharmaceuticals or lead compounds. The introduction provides almost no context on the state-of-the-art in thiolation. The authors state SN2 reactions have limitations, however these limitations can be overcome by SN1 processes, epoxide opening reactions, Mitsunobu reactions, or conjugate thiolations (among others). In this light, the introduction, framing of the work, and referencing should be significantly revised.

In terms of the merit of the work with respect to thiolation and decarboxylation methodology, my opinion is the work is not suitable for publication in Nature Communications. The scope of substrates examined is narrow. Poor yields for tertiary substrates are observed for non-biased substrates. The strategy for acid activation and capture is well established. Due to the technical nature of the advance this manuscript reports, it is better suited for publication in a journal such as Org. Lett. or Chem. Eur. J.

Response:

Regarding DOS, we have revised the introduction part and the corresponding references.

Regarding the background on thiolation, as our work is about decarboxylative thiolation, so we mainly focused on this aspect, and we discussed this background in the second paragraph as you might be aware. We also cited the related references (Ref. 62 is a book on thiol and thiolation; Ref. 45 is a review, and selected examples Ref. 46-53). Whereas, we have added three more references (Ref. 61, 64 and 65) in the revised manuscript. In particular, Ref. 64 by Schreiner (Org. Lett. 2006, 8, 1767) has a detailed discussion on the synthesis of tertiary thiols with a comparison.

The significance of developing decarboxylative transformation has been well appreciated by many famous group and journals: for a Review, see Ref. 25 Angew. Chem. Int. Ed. 54, 15632–15641 (2015).

For recent examples, see Ref. 26-41: Science 357, 283–286 (2017); Science 357, 283–286 (2017); Science 360, 419–422 (2018); Nat. Catal. 1, 120–126 (2018); Science 363, 1429–1434 (2019), etc.

Furthermore, the current transformation is about decarboxylative thiolation to free thiols, which, to the best of our known is a new reaction and also a new entry to decarboxylative transformation. The other two referees have also recognized the novelty of our work. Even in the sense of thiol synthesis, the using carboxylic acid as feedstock has its advantages (e.g. availability and diversity), when compared with alkyl halide, epoxide, etc. At least, different methods have their advantages and disadvantages. For example, the recent reported synthesis of 1-admantanethiol (Ref. 64 and 65) were performed under very harsh conditions (reflux in conc. aq. HBr/AcOH). Even under this conditions, only a trace amount of bicyclo[2.2.2]octane-1-thiol

was obtained. In contrast, with this decarboxylative method, we can prepared the tertiary bridgehead thiols (31, 33, & 34 in 52-60% yield) under mild conditions at room temperature.

Thank you for your time on evaluating this manuscript, suggestion and understanding.

Reviewer #3:

I In this work, Liao and coworkers developed a visible light-mediated direct decarboxylative thiolation reaction of alkyl redox active esters to free thiols is developed based on the abundant carboxylic acid feedstock, and the arylthioamides have been identified as an effective sulfur donor and crucial to this thiolation reaction. Importantly, this transformation provides a convenient and green channel for the preparation of free thiols and could be further in-situ diversification allows for a rapid and general access to various pharmaceutically compounds such as sulfide, disulfide, thiocyanide, and thioselenide. However, there are still some issues that need to be resolved before being published. Therefore, I recommend this manuscript publish in nature commucation after minor revised are needed.

1. For different activities of sulfur donors, the author should give corresponding explanations.

Response: In the beginning, the reaction mechanism was not clear (In fact, the mechanistic study took us almost one year). So we just screened different types of possible sulfur donors (Table 1, we shown some selected examples, and more sulfur donors were placed in Supplementary Table 1). Based on the mechanistic study and the mechanism depicted in Fig. 4, we have added some explanation and comments on the different activities of sulfur donors in the mechanistic discussion part. For details, please see the sentences highlighted in yellow on Page 8 & 9.

2. In Figure 2, the yields of some free thiols products are relatively low, what's the main byproduct?

Response: The main byproducts were the corresponding alkanes (R-H). We have added a comment on the results on Page 7 in the revised manuscript, with the examples of 35, 36, & 43.

3. If the carboxylic acid is used directly as the starting material, the corresponding product can be obtained by decarboxylation thiolation, or can the "one-pot" method be used to achieve this transformation?

Response: When we used the carboxylic acid as the starting material directly, the carboxylic acid cannot react under the standard conditions and no product was obtained. When we used the "one-pot" method, the decarboxylative thiolation can proceed, but the yield of the product is lower (59%). We have added this result in Supplementary Table 3 (entry 11).

4. Based on the speculated reaction mechanism, which step is the rate-determining step?

Response: Based on the TLC analysis and reaction optimization results, the transformation of imine intermediates (Int-C) to thiols could probably be the slow step, as after the starting redox active ester was completely consumed, the amount of thiols continued to increase. We have added a comment on this result in the end of Page 8 in the revised manuscript.

5. The "RSC Adv. 2016, 6, 70335" should be cited in mechanism proposed.

6. The "J. Org. Chem. 1962, 27, 93-95" should be cited in introduction.

Response: Thank you for the suggestion. We have cited these papers as Ref. 61 and Ref. 75 in the revised manuscript.

7. The figure S 7-9 should be provided the stern-volmer plots.

Response: We have added the corresponding stern-volmer plots after Figure S7-9 in the revised Supplementary Information (Figure S7-9). Thank you very much again for all the comments and suggestions.

REVIEWERS' COMMENTS

Reviewer #1 (Remarks to the Author):

Authors have addressed the points raised by the Reviewers, performed additional experiments and provided further explanations both in manuscript and SI. This reviewer recommend publication as it is.

Reviewer #3 (Remarks to the Author):

The author has answered all the reviewers' questions well in the revised manuscript. Therefore, I recommend accepting publication of the article on nature communications.